# Spatially-aware Photo-realistic Face Relighting using Joint Embedding of Light Properties

## Abstract

Single image face relighting is the challenging problem of estimating the illumination cast on images by a point light source varying in position, intensity and possibly colour. Learning the relationship between the light source properties and the face location is critical to the photo-realism of the estimated relit image. Prior works do not explicitly model this relationship which adversely affects the accuracy and photo-realism of the estimated relit image. We present a novel framework that explicitly models this relationship by integrating a novel light feature embedding with self-attention and cross attention layers in a custom image relighting network. Our proposed method estimates more photo-realistic relit images with accurate shadows and outperforms prior works despite being trained only on synthetic data. Our method is able to generalize to out-of-training light source positions and also achieves unsupervised adaptation from synthetic to real images.

## 1 Introduction

Single image face relighting is the complex problem of changing the illumination in the source image according to a given target light direction. It is an active area of research in computer vision and has applications in various domains such as face recognition (Le & Kakadiaris, 2019; Huang et al., 2020; Qing et al., 2004), background lighting transfer (Nestmeyer et al., 2020; Pandey et al., 2021) and image editing Li et al. (2018); Luan et al. (2017); Shih et al. (2014).

Two crucial aspects of face relighting are accurate rendering of shadows and preserving of facial details. A popular approach for single image face relighting is to rely on the estimated image intrinsics such as albedo and surface normal for relighting the image (Zhou et al., 2019; Hou et al., 2021; 2022). Additional information such as shadow masks (Hou et al., 2021) or face geometry (Hou et al., 2022) can be used to improve accuracy of estimated shadows. While these methods work well, the estimated relit images are not photo-realistic since the models have been trained on the inexact ground truth images in the DPR dataset (Zhou et al., 2019), and they estimate only the luminance channel & append colour channels from the input image. Some have tried to address these limitations by training on a new relighting dataset in a two-stage pipeline (Pidaparthy et al., 2024). While the shadows are sharp, they are often not photo-realistic.

None of the prior works explicitly addressed the fundamental challenge of image relighting, which is learning the complex relationship between light source information and face location & orientation. Ponglertnapakorn et al. (2023) tried to implicitly learn this relationship using the prior knowledge of large diffusion models, however their approach significantly increased the computation cost and inference time. Further, prior works only explore relighting with white colour light and lack a method to incorporate light colour.

To address these limitations, we designed a novel lighting embedding that allows joint modelling of the light source properties, which are the position, colour and intensity. We propose a novel lighting network that enables the model to learn the relationship between light source position, intensity and colour. This representative light feature is then combined with image features in a residual convolutional autoencoder. By integrating self-attention and cross-attention layers at multiple resolutions, the network allows for learning the relationship between light features and image features. These combined features are passed to a residual decoder which estimates the relit image. The pipeline

of our proposed image relighting network is shown in Fig 1. Through extensive experimental validation, we show that our method outperforms other SOTA methods and that our model generalizes very well to out-of-training light source properties. Additionally, we are able to achieve unsupervised adaptation from synthetic to real images.

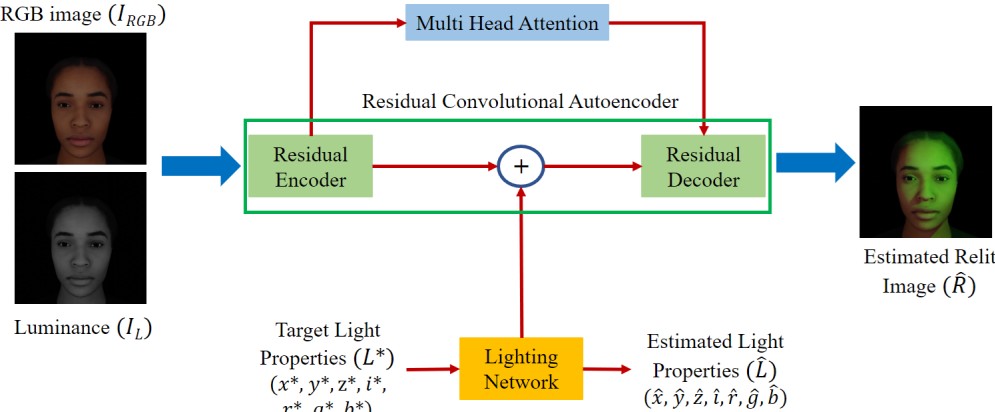

Figure 1: The proposed image relighting network architecture for face relighting is shown. The target light source properties are input to a lighting network which extracts light features. The RGB image and luminance channel are input to a residual encoder which learns the image features and combines them with the light features in a residual decoder to estimate the relit image. Multi-head attention layers are used to explicitly model the relationship between the light source and input image (face). Figure is best viewed in colour.

In summary, our contributions are:

- A novel light feature embedding that allows for joint modelling of the light properties (position, intensity and colour). This embedding provides an approach that can be easily extended to any other light properties.
- A novel lighting network that enables the network to explicitly learn the relationship between the light source position, intensity and colour.
- A novel image relighting network that explicitly models the relationship between the light source properties and face location & orientation.

## 2    PRIOR WORK

There are broadly four types of approaches explored for face relighting: 1) intrinsic image decomposition and rendering (Barron & Malik, 2014; Egger et al., 2018; Genova et al., 2018; Le & Kakadiaris, 2019; Lee & Lee, 2020; Lee et al., 2005; Li et al., 2014; Lin et al., 2020; Nestmeyer et al., 2020; Sengupta et al., 2018; Shahlaei & Blanz, 2015; Shu et al., 2017; Tewari et al., 2017; Tran et al., 2019; Tran & Liu, 2018; 2019; Wang et al., 2008; Yamaguchi et al., 2018), 2) image-to-image translation (Atoum et al., 2020; Tewari et al., 2021; Liu et al., 2021; Sun et al., 2019), 3) style transfer (Li et al., 2018; Luan et al., 2017; Shih et al., 2014; Shu et al., 2017; Pandey et al., 2021; Ponglertnapakorn et al., 2023; Yeh et al., 2022) and 4) ratio image estimation (Peers et al., 2007; Shashua & Riklin-Raviv, 2001; Wen et al., 2003; Stoschek, 2000).

Given an input image and light source position, the intrinsic image decomposition methods aim to estimate components such as albedo, surface normal, reflectance and lighting. These components are then used to render a relit image based on the light source position. The effectiveness of these methods heavily depends on the accuracy of each intrinsic map and a cascading of the estimation errors results in relit images that lack high frequency details and may contain artefacts.

To address the issue of cascading errors, some approaches have framed relighting as an image-to-image translation task. Although these methods can estimate relit images with varying accuracy (Sun et al., 2019; Zhou et al., 2019), they struggle with inaccurate shadow predictions and adapting to

different light intensities. Another approach to relighting is image style transfer (Li et al., 2018; Luan et al., 2017), where both source and references images are provided as inputs and the lighting of the reference image is transferred to the source image. Some have explored using environment lighting maps of a reference image to transfer lighting to a source image (Pandey et al., 2021; Ponglertnapakorn et al., 2023; Yeh et al., 2022). However, these approaches require high-quality non-occluded source and reference image pairs with diverse lighting variations, and ground truth environment lighting maps which can be expensive to obtain. In contrast, our method does not require any reference images or environment maps and can relight an image for any given light source position.

Some approaches have focused on learning a per-pixel multiplier map by estimating the ratio between the source and target images. However, these approaches require multiple input images (Peers et al., 2007; Shashua & Riklin-Raviv, 2001) or both source and target images (Stoschek, 2000) at inference, which limits their suitability for real-world applications.

Accurate estimation of shadows is critical to face relighting. Towards this end, some methods learn a weightage function on the estimated shadow mask (Hou et al., 2021), while some others have used ray-tracing to estimate the shadows approaches have explored the utility of a shadow pixels (Hou et al., 2022). While these methods estimate shadows with reasonable accuracy, the relit images often lack photo-realism due to hard boundaries in the shadow regions, whereas real shadows are softer and have diffused edges. Some have tried to overcome these limitations by improving the quality of the training dataset and combing attention features at multiple scales (Pidaparthy et al., 2024), however the estimated relit images has colour artefact issues and the performance suffers on input images with existing shadows. Another approach tried to leveraged the knowledge of large diffusion models (Ponglertnapakorn et al., 2023), but this method comes at a significant increase in computational costs and inference time. Also, several pre-processing blocks further increase the inference time.

We address several limitations of the prior work and propose a lightweight network architecture for fast inference on edge devices. We believe that incorporating light colour can improve the model performance. Hence, we design a novel light embedding that allows the model to learn the correlation between light source position, intensity and colour. Next, we design a novel network architecture that explicitly learns the relationship between the light source information and face location & orientation. This enables the model to accurately learn the face illumination and shadow strength, and thus, estimate a photo-realistic relit image.

## 3 DATASET

We obtained training and test datasets from Pidaparthy et al. (2024). We made a few important modifications to their dataset preparation strategy that significantly improved the quality of the ground truth dataset.

We believe that incorporating light colour information during training can improve the illumination on the face and reduce the colour artefacts in estimated relit images. Since there are no publicly available relighting datasets with coloured lighting, we generated our own dataset using the synthetic OLAT lighting rig in Blender software (Pidaparthy et al., 2024). Our dataset consisted of 7 maximally separated light colours: White $(255, 255, 255)$, Red $(255, 0, 0)$, Green $(0, 255, 0)$, Blue $(0, 0, 255)$, Yellow $(255, 255, 0)$, Magenta $(255, 0, 255)$ and Cyan $(0, 255, 255)$. Sample images from the training dataset are seen in Fig 2.

Next, we manually corrected the offset of the 3D models and aligned the face location all of models to be the same in the synthetic lighting rig. This ensured uniformity when augmenting the training dataset through rotation or displacement of the subject positions (3D models), which made it easier for the model to learn the relationship between light source information and face location & orientation. Additionally, we also modified the 3D models to have less reflective surfaces and cast sharper shadows. This significantly improved the photo-realism of the ground truth relit images.

The light source properties are represented as a 7D tuple of $(x, y, z, i, r, g, b)$ where $(x, y, z)$, $i$ and $(r, g, b)$ refers to the light position, intensity and colour, respectively. Using the same strategy as that in Pidaparthy et al. (2024), we sample the light position from a unit volume such that $x \in [-1, +1]$, $y \in [0.4, 1]$ and $z \in [-1, +1]$. The intensity $(i)$ is varied such that $i \in [0.4, 1]$. The $X$-$Z$ plane

is in the front of the 3D human model (face) and $Y$-direction indicates the frontal distance of the 3D human model from the light source. We generated 3,000 input-relit image pairs for each 3D human model while randomly varying the light source position, intensity and colour. We augmented the training dataset through left-right rotation of the 3D human models and changing the relative position between the 3D model and the camera.

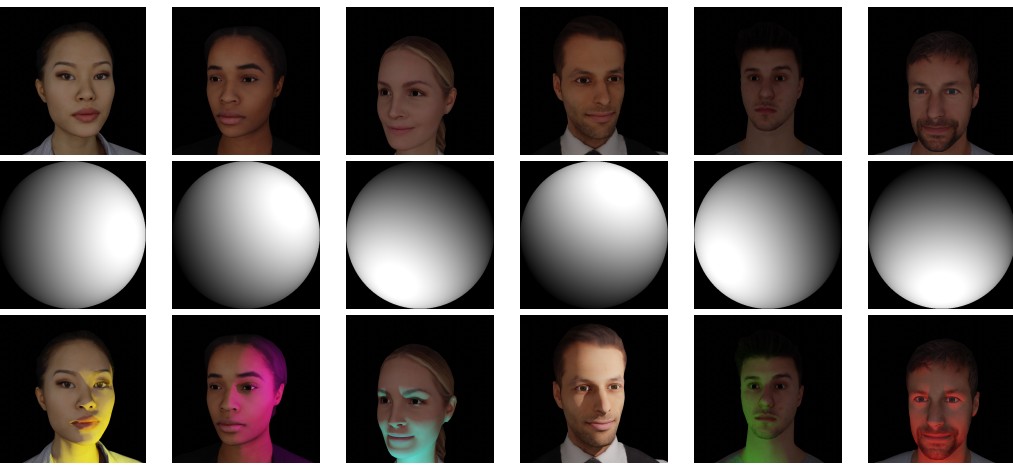

Figure 2: Sample images from the ground truth dataset. Top row: Input images; Middle row: Light source position; Bottom row: Ground truth relit images. Figure best viewed in colour.

# 4 IMAGE RELIGHTING NETWORK

A key challenge in face relighting is precisely estimating the illumination and shadows cast on the face by a light source. This involves learning the complex relationship between the face location & orientation and the light source properties. In this section, we describe our novel lightweight image relighting network (Fig1) that achieves two goals: 1) accurate estimation of shadows and illumination on the face and 2) efficient inference on edge devices. To achieve this, we propose a novel light embedding that jointly models the light source properties. We train a lighting network that learns the correlation between the light source position, intensity and colour. These embeddings are combined with image features, learnt using a residual convolutional autoencoder, in cross-attention blocks that capture the relationship between light and face information. The combined features are then passed to a decoder to generate a photo-realistic relit image.

## 4.1 LIGHTING NETWORK

Most prior work encoded only the light source position using a 9-dimensional Spherical Harmonics (SH) vector, which captures the maximum variance of the light source position moving along a unit sphere (Zhou et al., 2019; Hou et al., 2021; 2022; Ponglertnapakorn et al., 2023). More recent works have appended light source intensity to obtain a 10-D SH vector (Pidaparthy et al., 2024). The SH vector is passed to a lighting network to obtain the light features.

Although SH vectors are helpful for handling variations in the light positions across a sphere, they have some limitations. The SH vector models only the light source position. Even when intensity is appended, they fail to capture the correlation between the light source position and intensity. The illumination on an image depends on both position and intensity since a high-intensity light farther from the face and low-intensity light closer to the face might cast similar illuminations. Our initial experiments showed that the SH vector might not be adequate to model the wider range of variations in light source positions and intensities. Additionally, SH vector does not account for light colour, which we believe is crucial for improving the accuracy of illumination on the face and reducing the colour artefacts. This necessitates a more comprehensive light embedding that jointly models the light source properties and provides a rich representation of the correlation between them.

Towards this end, we propose a more complex representation for the 7D light source tuple that can be easily extended to any other properties of the light source. Our proposed encoding for the position, intensity and colour of the light is inspired by the positional encoding used in vision transformers (Dosovitskiy et al., 2020; Vaswani et al., 2017).

Let $l = (x, y, z, i, r, g, b)$ be the target light source 7D tuple. The $x$-position is encoded by a vector of dimension $d = 128$, which we denote as $L_x$. The components of $L_x$ are

$$L_x(2j) = \sin\left(\frac{x}{n^{2j/d}}\right) \quad \text{and} \quad L_x(2j + 1) = \cos\left(\frac{x}{n^{2j/d}}\right) \tag{1}$$

where $n = 10,000$ is a scaling factor and $j$ is the index position of the array $L_x$ such that $0 \leq j \leq d/2 - 1$. Odd positions are represented by a sine function and even positions by cosine function.

In a similar manner, we compute the embeddings for the other 6 components which we denote as $L_y$, $L_z$, $L_i$, $L_r$, $L_g$ and $L_b$ (Fig 3a). We concatenate the seven embeddings to obtain an 896-dimensional light embedding, $L^* = [L_x \ L_y \ L_z \ L_i \ L_r \ L_g \ L_b]$. Instead of using a Multi Layer Perceptron (MLP) for computing the light features like prior works have done, we use a convolutional network that reduces the number of computations and optimizes the inference time on edge devices. We zero-pad the light embedding ($L^*$) and reshape them to a $1 \times 32 \times 32$ image. This image is passed to a lighting network (Fig 3(b)) which consists of an encoder and decoder module both having three convolution layers with 32, 128 and 512 channels. After each convolution layer, Multi DConv Head Attention (MDHA) modules ((Pidaparthy et al., 2024)) with self-attention layers[1] are used. This enables the network to learn the correlation between the light source position, intensity and colour. The output of the encoder is a $128 \times 16 \times 16$ image which is passed through an MDHA module to obtain the light features ($X_L$) that are combined with image features in a residual convolutional autoencoder.

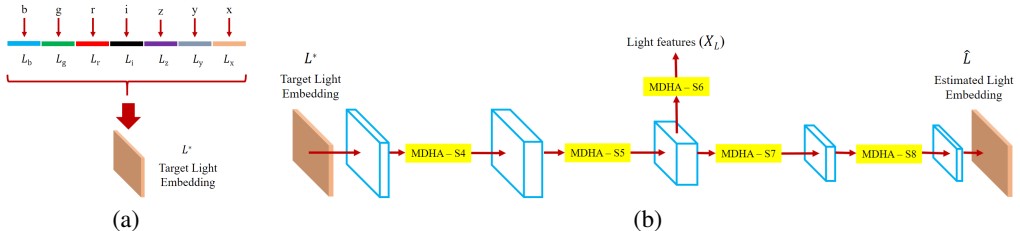

Figure 3: (a) For each component of target light 7D tuple, we compute a novel light embedding which is then concatenated and reshaped to obtain the target light embedding ($L^*$). (b) The proposed light network consists of an encoder and decoder, each having three convolutional layers (indicated in blue) with a MDHA module is used after each convolutional layer. The light features ($X_L$) is computed on the output of the encoder module.

## 4.2 RESIDUAL CONVOLUTIONAL AUTOENCODER

We trained a modified ResNet-34 based residual convolutional autoencoder to estimate the relit image (Fig 4). The input data ($I_D$) has 4 channels: RGB image ($I_{RGB}$) and luminance channel ($I_L$). The input data is passed to the encoder which consists of four ResNet blocks having 16, 32, 64 and 128 channels, respectively. The feature maps are downscaled by half after each block. The 128-D image features ($X_I$) output by the encoder are added with the light features ($X_L$) as seen in Fig 4. The combined output is then passed through a decoder that estimates the relit image. The decoder is a mirror of the encoder and consists of four ResNet blocks having 128, 64, 32 and 16 channels, respectively, and the feature maps are upscaled by a factor of 2 after each block. Multi DConv Head Attention (MDHA) modules (Pidaparthy et al., 2024) have been used at each level of the skip connection since they help preserve fine-grained facial details.

The two main goals of the residual convolutional autoencoder are: 1) learn the relationship between face location & orientation and light source information; and 2) estimate accurate and photo-realistic relit images. None of the prior works tried to explicitly learn the rich and dense relationship between the light source information and the face location & orientation. We propose to use cross-attention

---

[1] These modules are indicated as MDHA-S4, S5, S6, S7 and S8.

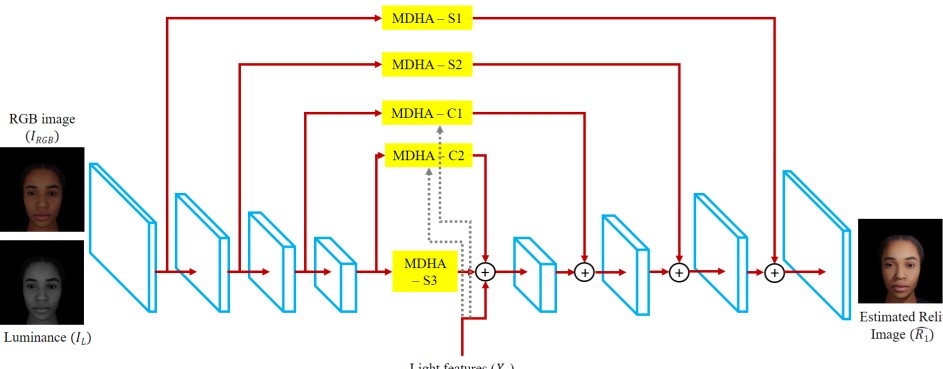

Figure 4: The architecture of the proposed residual convolutional autoencoder is shown. The encoder and decoder consists of four ResNet blocks each (indicated in blue). Five MDHA modules are used at each level of the skip connections between the encoder and decoder. The light features are passed to the cross-attention layers of two MDHA modules (indicated with grey lines).

layers in the MDHA modules to learn the relationship between the light source information and face location, and then leverage this understanding at higher resolutions of the decoder to improve the accuracy and photo-realism of the estimated relit image.

Recall that the deeper layers in a deep neural network (lower resolution layers) learn high-level image features, while the earlier layers in the network (high resolution layers) learn low-level images features. Thus, to explicitly learn the complex relationship between light source properties and face location & orientation, we propose to use cross-attention layers in the MDHA modules at lower resolutions as seen in Fig 4[2]. The output of these MDHA modules are the composite features (obtained by combining the image and light features) which are computed as

$$Q = W_q\, X_I \quad K = W_k\, X_L \quad V = W_v\, X_L \quad X_C = \text{softmax}\left(\frac{QK^T}{\sqrt{d_k}}\right)V \quad (2)$$

where $Q$, $K$, $V$ denote the query, key and value matrices, respectively (Vaswani et al., 2017; Dosovitskiy et al., 2020). $W_q$, $W_k$, $W_v$ denote the learned weight matrices and $X_I$, $X_L$ denote the image and light features. $X_C$ denotes the composite feature representation obtained by combining the light and image features. We estimate the query from the image features and the key-value pairs from the light features.

The output features of the "MDHA-C1" and "MDHA-C2" layers are the passed to the higher resolution layers for refining the fine-grained facial details. Hence, self-attention layers are used in the MDHA modules at higher resolution (see "MDHA-S1" and "MDHA-S2" in Fig 4). This ensures that the decoder estimates a more photo-realistic image with accurate shadows and illumination.

### 4.3 TRAINING LOSSES

We used three different losses for training the image relighting network: 1) lighting loss, 2) image reconstruction loss and 3) VGG loss.

All prior works have trained the lighting network to optimize the MSE loss on the 9-D or 10-D SH vector. However, MSE is sensitive to the magnitude of the vectors and hence, it can be misleading when calculating the distance between two vectors. Further, changes in components of the light vector have more effect on the output results than other components. MSE might not be able to accurately capture this. Hence, we used the cosine dissimilarity metric to measure the distance between the estimated and ground truth light embeddings.

Let $L^*$ be the light embedding for the target light source 7D tuple (ground truth). Let $\hat{L} = \mathcal{N}_\mathcal{L}(L^*)$ be the estimated light embedding, where $\mathcal{N}_\mathcal{L}$ refers to the lighting network. We define the light loss

---

[2]"MDHA - S1' and "MDHA - C1" denote to the MDHA modules with self-attention layers and cross-attention layers.

as

$$L_{light} = 1.0 - \frac{L^* \cdot \hat{L}}{||L^*|| \, ||\hat{L}||} \tag{3}$$

We measured the image reconstruction loss at both global ($L_{global}$) and local ($L_{local}$) scales. The image reconstruction loss had two components: 1) structural dissimilarity (DSSIM) loss ($L_{dssim}$) and 2) smooth L1 loss ($L_{smooth}$). We define DSSIM loss as $L_{dssim} = \frac{1-SSIM}{2}$, where $SSIM$ score is computed using the built-in PyTorch function to compute the $SSIM$ score.

We define the smooth L1 loss as

$$L_{smooth}(R^*, \hat{R}) = \begin{cases} (R^* - \hat{R})^2 & \text{if } |R^* - \hat{R}_1| < 0.5, \\ |R^* - \hat{R}| - 0.25 & \text{otherwise} \end{cases} \tag{4}$$

where $R^*$ and $\hat{R} = \mathcal{N}_{\mathcal{R}}(I_D)$ are the ground truth and estimated relit images, respectively. $N_R$ and $I_D$ refer to the residual convolutional autoencoder and the 4-channel input data, respectively. To compute the loss at local scales, we divided the image into $128 \times 128$ pixel patches, overlapping by 50%. Let $P^*$ and $\hat{P}$ be the image patches from the ground truth and estimated relit images, respectively. We define the global and local image reconstruction losses as

$$L_{global} = \lambda_1 \, L_{smooth}(R^*, \hat{R}) + \lambda_2 \, L_{dssim}(R^*, \hat{R}) \tag{5}$$

$$L_{local} = \sum_k \lambda_3 \, L_{smooth}(P^*, \hat{P}_1) + \lambda_4 \, L_{dssim}(P^*, \hat{P}) \tag{6}$$

where $k$ is the total number of patches.

We also computed the VGG loss ($L_{vgg}$) using the output of the first three convolutional blocks of the pre-trained VGG-19 network (Simonyan & Zisserman, 2014) ($\mathcal{N}_{vgg}$). The VGG loss is defined as

$$L_{vgg} = 1.0 * ||\mathcal{N}_{vgg_{b1}}(R^*) - \mathcal{N}_{vgg_{b1}}(\hat{R})||_2^2 \; + \; 0.8 * ||\mathcal{N}_{vgg_{b2}}(R^*) - \mathcal{N}_{vgg_{b2}}(\hat{R})||_2^2$$
$$+ \; 0.6 * ||\mathcal{N}_{vgg_{b3}}(R^*) - \mathcal{N}_{vgg_{b3}}(\hat{R})||_2^2 \tag{7}$$

where $\mathcal{N}_{vgg_{b1}}$, $\mathcal{N}_{vgg_{b2}}$ and $\mathcal{N}_{vgg_{b3}}$ are the output of the first, second and third convolutional blocks.

Thus, the final loss used for training the image relighting network was

$$L_{final} = L_{global} \; + \; L_{local} \; + \; \lambda_5 \, L_{light} \; + \; \lambda_6 \, L_{vgg} \tag{8}$$

where $\lambda_1 = \lambda_5 = 1$, $\lambda_2 = \lambda_3 = \lambda_6 = 10$ and $\lambda_4 = 100$ are the weights for each loss term.

## 4.4 TRAINING DETAILS

Unlike prior works, we do not estimate any image instrinsic map nor do we use any additional networks during training. Our single-stage pipeline is able to estimate photo-realistic relit images with high accuracy and achieve unsupervised domain adaptation from synthetic to real images. We achieve this using specific data augmentation techniques while preparing the training dataset.

We generating the training dataset, we randomly apply left-right rotation on the 3D model of $r \in [-60, +60]$ degrees. We also vary the position of the 3D model w.r.t the camera by a displacement of $d_x \in [-0.3, +0.3]$ and $d_z \in [-0.3, +0.3]$ along $X$ and $Z$ directions, respectively. Further, we used two different types of data augmentation during training: 1) image flipping and 2) brightness and contrast jitter. The image is flipped horizontally (left-right mirroring) and the light source position is appropriately updated. The brightness and contrast of the RGB image are adjusted by an additive factor of $b$ and multiplicative factor of $c$, where $b \in [-20, +20]$ and $c \in [0.8, 1.2]$. These augmentations result in a training dataset that captures multiple different variations of the input images and thus, improves the generalization of the model to real images.

The image relighting network was trained on a dataset of 21,000 images and validated on 3,000 images. The input data to the residual convolutional autoencoder consisted of 4-channels - RGB image ($I_{RGB}$) and luminance channel ($I_L$), which were resized to $512 \times 512$ pixels. The input to the lighting network was the light features embedding computed on the target light source 4D tuple.

The training loss was optimized using Adam optimizer (Kingma & Ba, 2014) with L2 regularization of $0.01$. The initial learning rate of $1e-5$ was decayed by a factor of $0.9$ after each epoch and the network was trained for 30 epochs with a batch size of $8$. In a manner similar to all prior works, we also work with the segmented foreground region. We used the segmentation masks obtained using the pre-trained Mask R-CNN (He et al., 2017) model in PyTorch.

## 5 RESULTS

We discuss the quantitative and qualitative performance of our model in this section. We evaluated the quantitative performance of the model on two real image test datasets provided by Pidaparthy et al. (2024): 1) Multi-Pie (MP) test dataset and 2) Real Human (RH) test dataset. The Multi-pie dataset (Gross et al., 2010) has 6,474 images, captured using 249 subjects with 2 different expressions and 13 different light source positions. The Real human dataset consisted of 432 images with 6 different subjects and 72 images per subject. This dataset consisted of significantly different out-of-training light source positions, which tested the generalization capabilities of the models. For a fair comparison with prior works, we evaluated our model on these datasets that used white coloured light.

We compared the performance of our model against four prior works (Zhou et al., 2019; Hou et al., 2021; 2022; Pidaparthy et al., 2024). All these methods estimated the relit image given a single input image and the target light source position[3]. Additionally, they are also lightweight for fast inference on edge devices. We quantified the performance using three metrics: 1) MSE, 2) DSSIM[4] and 3) LPIPS (Zhang et al., 2018). Both DSSIM and LPIPS have been shown to be highly correlated with the perceptual quality of the images (Nestmeyer et al., 2020; Zhang et al., 2018). The quantitative results can be seen in Table 1.

| Model | Trained on real images | # training examples | Dataset | MSE ↓ | DSSIM ↓ | LPIPS ↓ |
|---|---|---|---|---|---|---|
| Zhou et al. (2019) | ✓ | 135,000 | RH | 0.0716 | 0.2988 | 0.3736 |
| Hou et al. (2021) | ✓ | 180,000 | RH | 0.0090 | 0.1906 | 0.2650 |
| Hou et al. (2022) | ✓ | 180,000 | RH | 0.0152 | 0.0787 | 0.1522 |
| Pidaparthy et al. (2024) | ✗ | 21,000 | RH | 0.0049 | 0.0336 | 0.0741 |
| Ours | ✗ | 21,000 | RH | **0.0043** | **0.0307** | **0.0701** |
| Zhou et al. (2019) | ✓ | 135,000 | MP | 0.0845 | 0.3548 | 0.4389 |
| Hou et al. (2021) | ✓ | 180,000 | MP | 0.0125 | 0.2801 | 0.2538 |
| Hou et al. (2022) | ✓ | 180,000 | MP | 0.0118 | 0.2850 | 0.2607 |
| Pidaparthy et al. (2024) | ✗ | 21,000 | MP | 0.0096 | 0.0639 | 0.1361 |
| Ours | ✗ | 21,000 | MP | **0.0079** | **0.0587** | **0.1323** |

Table 1: Performance comparison of our model against prior works on two real image test datasets: 1) Real Human test dataset (RH) and 2) Multi-Pie dataset (MP).

Our model outperforms all the prior works on both real image test datasets. All the metrics are substantially lower than the prior works, indicating that our estimated relit images are much more accurate and photo-realistic. Despite training on multiple different light colours as compared to prior work, our method is able to outperform all prior works on white light. We used a similar training dataset to that used by Pidaparthy et al. (2024)[5], and our model significantly outperforms their work, especially on the challenging Multi-pie (MP) dataset. This shows the benefits of our design choices which include a novel light embedding and a network architecture that explicitly learns the relationship between the light source information and face location & orientation. As the real human (RH) test dataset consists of out-of-training light source positions, the metrics indicate that our model generalizes significantly better than prior works to these new light positions.

The qualitative results shown in Fig 5 backs up the findings in Table 1. We observed that the illumination on the face and accuracy of the shadows are significantly better than prior works. In

---

[3]All prior works used only $(x, y, z)$ except Pidaparthy et al. (2024) who used $(x, y, z, i)$.

[4]We compute DSSIM as DSSIM = $\frac{1-SSIM}{2}$ where SSIM measures the structural similarity.

[5]We mainly added the light colour as an additional component.

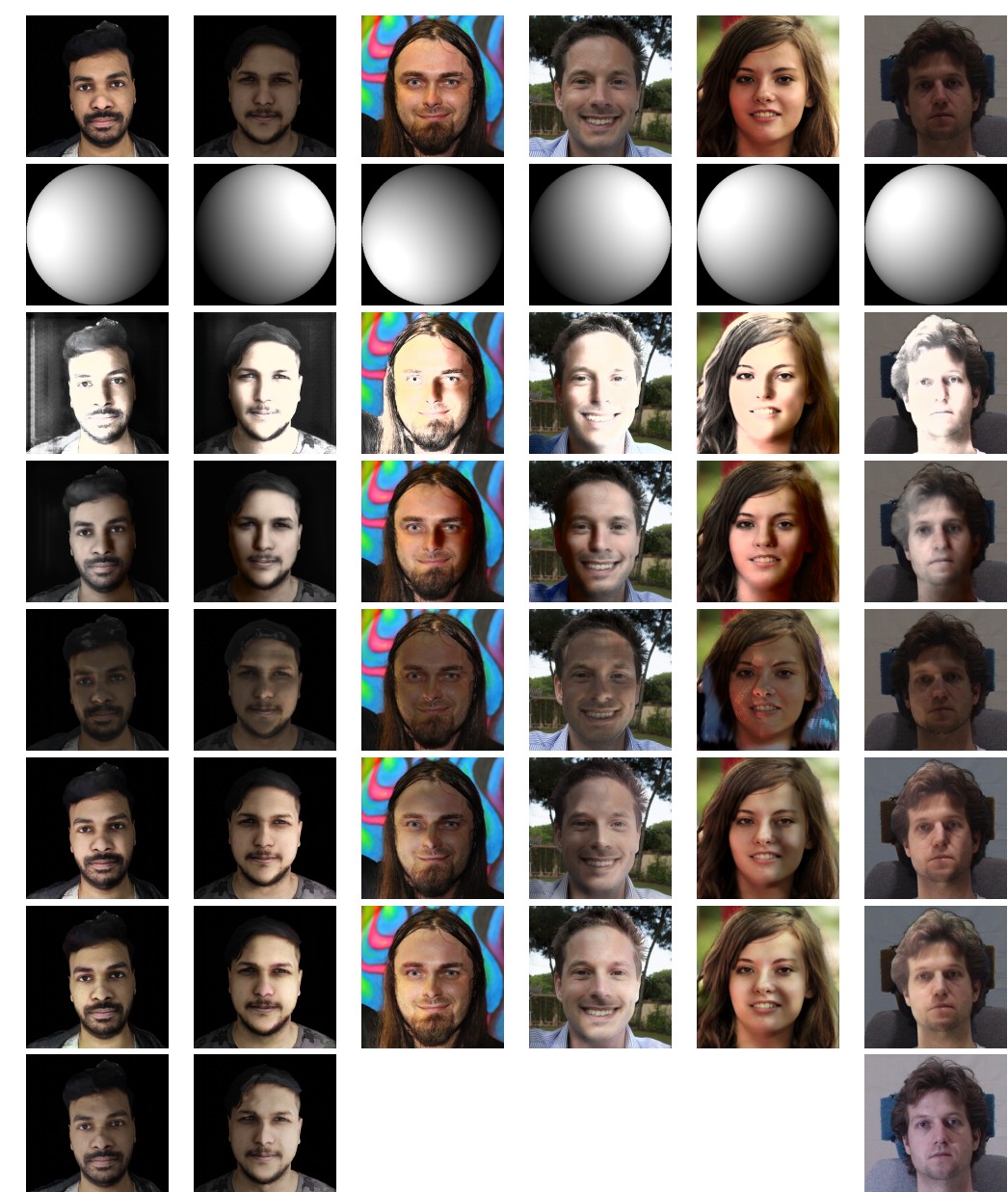

Figure 5: Qualitative comparison of our model against other methods on the real human test dataset (RH) (columns 1, 2), Celeb-FFHQ dataset (columns 3-5) and Multi-pie dataset (MP) (last column). Rows 1 and 2 are the input image and light source position; rows 3-6 are the results from Zhou et al. (2019), Hou et al. (2021), Hou et al. (2022) and Pidaparthy et al. (2024); rows 7 and 8 are our results and ground truth relit images. We do not have ground truth relit images on the Celeb-FFHQ dataset. Images are best viewed in colour.

many of the prior works, the shadows are incorrectly cast across the forehead. Our method is able to illuminate the face correctly. We also observed that the estimated relit images from our method have significantly improved the colour artefacts issues observed in prior works (see Fig 5 columns 3,4,6). This shows the benefits of modelling the light colour in addition to position and intensity. Our method is also able to handle directional lighting and shadows on the input images much better than prior works (see Fig 5 column 3, 5) since we specifically learn the relationship between input features and light features.

Fig 6 shows the generalization of our model across different light source colours. We observe that the model is able to apply light colour appropriately and generate accurate shadows (see Fig 6 columns 1 and 2). Our method is able to generalize across a variety of input images that have different ambient lighting and directional shadows, and different facial structures & ethnicities.

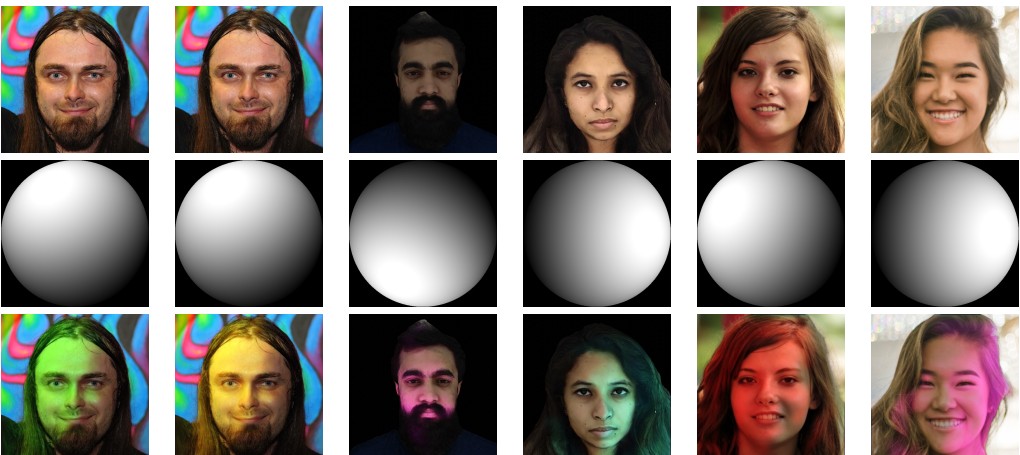

Figure 6: Qualitative results from our model for different colours of target light source. Top row: input images; middle row; light source position and last row: predicted results.

## 6  LIMITATIONS FUTURE WORK

We have shown the effectiveness of our methods for accurately modelling shadows and generating photo-realistic relit images. However, there are some limitations of the model. Currently, a segmentation mask is used at inference to segment the foreground object (human) which is passed as input to the model. One possible future extension of this work is to automatically localize the face in the input image and relight the image without using any foreground segmentation mask. Another possible extension is to capture the effect of multiple light sources by modelling their effect as a single composite light source.

## 7  CONCLUSION

We proposed a novel approach for face relighting given a single image and a light source position. We used a novel light embedding that jointly modelled the light source position, intensity and colour. The network enables learning the correlation between these parameters. We used cross-attention layers in the convolutional autoencoder to explicitly learn the relationship between the light source properties and the face location. Qualitative results show the benefit of various design choices and our model generates photo-realistic relit images. Our model is also able to easily generalize across multiple different coloured lighting. Quantitative analysis showed that our model outperforms SOTA methods on two challenging real image datasets. Our model is lightweight and has only 9.4M parameters.

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
