# SPATIALLY-AWARE PHOTO-REALISTIC FACE RELIGHTING USING JOINT EMBEDDING OF LIGHT PROPERTIES

## 1 ABLATION STUDY

We performed an ablation study to show the benefits of our network design choices. We replaced specific modules or inputs and retained everything else. This enabled us to evaluate the performance drop-off without specific modules. We evaluated the following design choices

1. Using SH coordinates for light feature embedding[1] instead of our novel light feature embedding
2. Not using any MDHA modules
3. Not using MDHA modules with cross-attention layers
4. Using only global losses ($\lambda_3, \lambda_4 = 0$)
5. Using only local losses ($\lambda_1, \lambda_2, \lambda_6 = 0$)

We evaluated the performance of the model with various ablations on the real human test dataset (RH). The quantitative results are shown in Table 1. We observed that using the MDHA modules (transformer blocks) are very critical to model performance. Cross-attention layers are also important to improve the visual quality of the final results. Using SH coordinates as light feature embedding worsens the model performance and global loss improves model performance more than local loss. The best performance was observed by the full model using all our design choices.

We observed that the performance of the models in Table 1 was better than the prior works in Table 1 of the main paper. This could be due to the training dataset strategy[2] which enables training a model with good performance despite removing important modules in the network architecture.

| Novel light embedding | MDHA modules | Cross-attention layers | Global loss | Local loss | MSE ↓ | DSSIM ↓ | LPIPS ↓ |
|:---:|:---:|:---:|:---:|:---:|:---:|:---:|:---:|
| ✗ | ✓ | ✓ | ✓ | ✓ | 0.0055 | 0.0367 | 0.0732 |
| ✓ | ✗ | ✗ | ✓ | ✓ | 0.0111 | 0.0542 | 0.0868 |
| ✓ | ✓ | ✗ | ✓ | ✓ | 0.0049 | 0.0361 | 0.0729 |
| ✓ | ✓ | ✓ | ✗ | ✓ | 0.0046 | 0.0338 | 0.0723 |
| ✓ | ✓ | ✓ | ✓ | ✗ | 0.0061 | 0.0369 | 0.0733 |
| ✗ | ✗ | ✗ | ✓ | ✓ | 0.0133 | 0.0742 | 0.1243 |
| ✓ | ✓ | ✗ | ✗ | ✓ | 0.0122 | 0.0681 | 0.1045 |
| ✓ | ✗ | ✗ | ✓ | ✗ | 0.0141 | 0.0784 | 0.1468 |
| ✓ | ✓ | ✓ | ✓ | ✓ | **0.0043** | **0.0307** | **0.0701** |

Table 1: Ablation study to evaluate the benefit of various design choices for the image relighting network.

## 2 ADDITIONAL COMPARISON AGAINST PRIOR WORKS

Fig 1 shows some additional comparison results against prior works. We observe that the results from our model are significantly more accurate and photo-realistic than prior works. Our method is

---

[1]The lighting network was trained to estimate the 9-dimensional SH coordinates.

[2]We mostly used a similar strategy to that proposed by Pidaparthy et al. (2024), but made a few important modifications as discussed in the main paper.

able to easily handle variations in input image such as changes to ambient lighting, facial structure and ethnicities, directional shadows in the input image, etc.

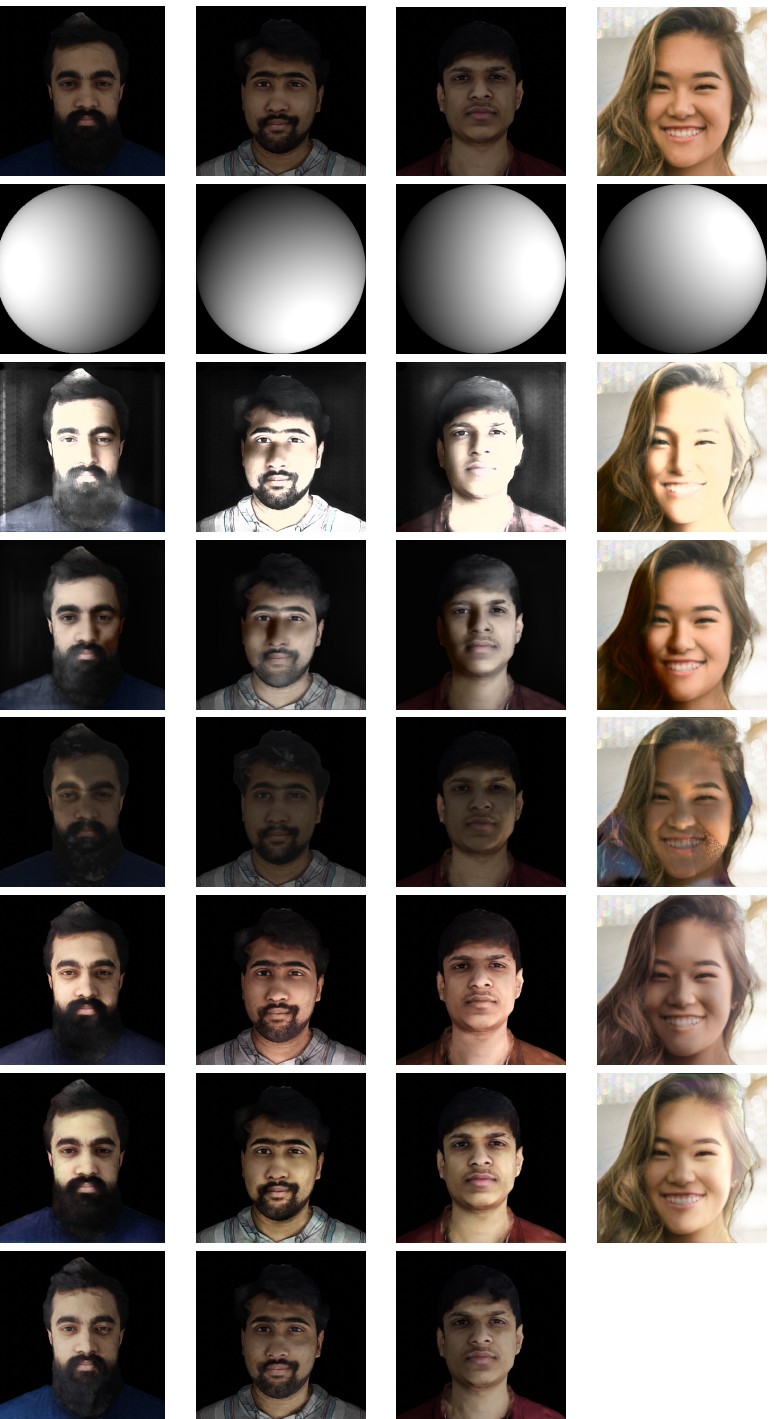

Figure 1: Qualitative comparison of our model against other methods on the real human test dataset (RH) (columns 1, 2), Celeb-FFHQ dataset (columns 3-5) and Multi-pie dataset (MP) (last column). Rows 1 and 2 are the input image and light source position; rows 3-6 are the results from Zhou et al. (2019), Hou et al. (2021), Hou et al. (2022) and Pidaparthy et al. (2024); rows 7 and 8 are our results and ground truth relit images. We do not have ground truth relit images on the Celeb-FFHQ dataset. Images are best viewed in colour.