# OpenReview forum: "Spatially-aware Photo-realistic Face Relighting using Joint Embedding of Light Properties"
_ICLR.cc/2025/Conference — ICLR 2025 Conference Withdrawn Submission_

### Official Review · Reviewer_xt6i · 2024-10-27

**Soundness:** 3
**Presentation:** 3
**Contribution:** 2
**Rating:** 5
**Confidence:** 2

**Summary:**

This paper addresses the problem of single-image face relighting by generating images illuminated by a point light source with variable position, intensity, and potentially color. The proposed framework explicitly models the relationship between light source properties and face orientation by integrating a light feature embedding within self-attention and cross-attention layers in an image relighting network. Trained solely on synthetic data, the method produces relit images with shadows that outperform previous approaches. It demonstrates generalization to out-of-training light source positions and achieves unsupervised adaptation from synthetic to real images.

**Strengths:**

This paper introduces a lighting embedding that enables joint modeling of light source properties, specifically position, color, and intensity. The image relighting network explicitly models the relationship between light source characteristics and the face’s spatial configuration. Experimental results indicate that the proposed method surpasses existing techniques on two benchmark datasets.

**Weaknesses:**

Overall, I think the proposed method lacks novelty, and the experimental results are unconvincing.

- Utilizing a 7D vector (position, color, intensity) to represent point light sources is a basic form of lighting representation. Applying positional embedding (PE) to this 7D vector for lighting embedding is not novel, as PE for high-frequency encoding is common in transformers and neural radiance fields (NeRFs). The network architecture is a simple encoder-decoder, and the use of cross-attention and self-attention layers in the relighting network is also straightforward. The design does not contribute new insights to the field.
- The evaluation is insufficient. The relighting results are hard to assess fully, as no video sequences are provided to demonstrate images illuminated under rotated lighting. The visual results are limited and unsatisfactory, and the quantitative improvements over Pidaparthy et al. (2024) are limited.
- The synthetic-to-real adaptation could be validated more robustly with additional examples across diverse real images and lighting conditions, as current qualitative results do not sufficiently demonstrate robustness in varied real-world scenarios.
- Claims regarding edge device optimization remain unsupported by runtime benchmarks.

Minor:
- Certain visual figures (e.g., Fig. 5) could benefit from clearer labeling to improve the clarity of comparisons with prior methods.

**Questions:**

- Clarification on the novelty of the proposed approach
- Lack of sufficient evaluation
- Discussion of runtime

Refer to weaknesses for details.

---

### Official Review · Reviewer_gxNc · 2024-10-29

**Soundness:** 2
**Presentation:** 2
**Contribution:** 1
**Rating:** 1
**Confidence:** 4

**Summary:**

The paper presents a face-relighting method. The method takes in a reference image under another lighting with the image’s luminance and outputs the relit image under the target lighting. The authors claim that using colored OLAT images can enhance performance and prepare a dataset with seven fixed colors. A new lighting network that consists of CNN modules and MDHA modules is also proposed to convert the lighting condition (including the point lighting’s intensity, position, and color) into a high-dimension feature. This feature is then input with the other inputs into the residual convolutional autoencoder to output the final relit image. Evaluations show that the proposed method outperforms sota methods.

**Strengths:**

1. The paper adds lighting color into the lighting embedding with a new lighting encoder to [Pidaparthy et al., 2024] and achieves higher metrics on test datasets.

**Weaknesses:**

1. The proposed method is very incremental to [Pidaparthy et al. (2024)]. The overall pipeline (learned lighting encoding + residual conv ae for relit image generation) is unchanged. And the residual convolutional autoencoder component is the same. Thus, the novelty is low.
2. If efficiency is the goal (can run on edge devices), it should be explicitly marked in the title, abstract, and introduction. The inference speed should also be compared with other methods, e.g. inference time per image, model size and even power-consumption during inference.
3. The data quality is very low.
    1. The 3D models in the collected dataset do not have important PBR information for realistic portrait relighting, e.g. accurate specular modeling and subsurface scattering. The quality of the referenced paper for data source [Pidaparthy et al. (2024)] is below the bar of top conferences like ICLR.
    2. The color of the lighting in the training data is not continuous, why use separated colors instead of continuously sampled ones? Sampling only the maximally separated ones can cause problems in interpolation. Please justify this.
4. I don’t see the connection between adding lighting color and enhancing accurate shadow or help learning the relationship between the light source properties and the face location. Shadow/visibility has nothing to do with lighting color but only with lighting positions.
5. L213: ‘SH does not account for model light color’: you can concatenate the lighting color with SH coefficients.

Typo: L147: artifacts?

**Questions:**

1. In section 6 it is mentioned that a segmentation mask is provided to the model, but this input is not mentioned in section 4. Do you input a segmentation mask or not?
2. Supplement ablation variant 1, do you provide lighting color here?

---

### Official Review · Reviewer_ApxL · 2024-10-31

**Soundness:** 2
**Presentation:** 2
**Contribution:** 2
**Rating:** 3
**Confidence:** 5

**Summary:**

This submission presents a novel method to model the relationship between lighting attributes (color, intensity, position) and the face itself (orientation, semantic location) in an attempt to improve the relighting performance. This is done by using a lighting network as well as a convolutional autoencoder combined with multi-head self and cross attention to model the relationships between face features and lighting features. Experiments demonstrate state-of-the-art performance on two datasets quantitatively and qualitatively: Multi-PIE (a controlled dataset) and Real Human (out of training distribution lighting conditions). The method can also handle different light colors, which is largely absent from relighting methods that do not leverage real captured light stage data and environment maps.

**Strengths:**

State of the art performance quantitatively on Multi-PIE and Real Human compared with several existing baselines.

The ability to handle different light colors is absent from many methods trained without real light stage data and environment maps.

**Weaknesses:**

There is a complete lack of ablations in the paper. Minimally, a natural ablation that I would expect is whether the cross attention between image features and lighting features actually leads to a quantitative and qualitative improvement in performance. It is otherwise hard to gauge the significance of the claimed contributions.

On a related note, the second contribution about modeling the relationship between light color, intensity, and position is difficult to accept given the environment map exists as a lighting representation and models all three components. If the authors wish to highlight that they don't require environment maps or light stage data during training, this should be reflected in the contributions to avoid confusing readers.

In the experiments section, why is there no comparison with the DiFaReli: Diffusion Face Relighting method? It is one of the most recent in-the-wild face relighting methods. The authors claim that it adds additional inference time but there are no experiments in the paper or claims to novelty related to inference time. Thus, it should be compared against quantitatively and qualitatively.

Qualitatively, the relighting results of this work do not strike me as noticeably better than prior work, especially compared with Pidaparthy et al. (2024). To me, many of the images (even with ground truth) seem to have comparable quality or it's unclear from the provided information which is better.

There are some mistakes in the paper. For example, the methods of Hou et al. 2021 and 2022 do not use the same dataset: please examine this carefully. There are errors related to this both in Table 1 and the introduction.

There is also almost no detail about the Real Human dataset except that it contains out of distribution lighting conditions. It would be better to be more specific so that readers are more convinced of the comprehensiveness of evaluations.

There are several important citations missing in this work from the relighting domain:

-COMPOSE: Comprehensive Portrait Shadow Editing (ECCV 2024)

-SwitchLight: Co-design of Physics-driven Architecture and Pre-training Framework for Human Portrait Relighting (CVPR 2024)

-NeRFFaceLighting (SIGGRAPH 2023)

**Questions:**

I would suggest rewording or rethinking what the contributions in this work are. As it stands, I am not convinced that the claimed contributions are valid (e.g. second contribution about modeling relationships between lighting attributes when environment maps exist). Please check carefully whether among face relighting methods that do not require environment maps, something similar has been done. If not, reword this contribution to explicitly mention that it is novel w.r.t methods that do not require env maps and real captured light stage data.

I would also suggest creating an ablation study table to better convince readers of the paper's contributions. As I mentioned in Weaknesses, I'd minimally expect an ablation with and without the cross attention layers between image and lighting features. In addition, experiments against additional recent baselines such as DiFaReli would be appreciated. This includes quantitative and qualitative comparisons. For quantitative, we can use the metrics presented in Table 1. For qualitative, I'd like to see the results on the images presented in Figure 5. As of now, the only recent baseline is Pidaparthy et al. (2024).

Please include more information about the Real Human dataset since this is lacking in the submission. Ideally we should discuss more about the range of lighting conditions, types of poses/expressions included and any additional augmentations found in the dataset. Simply saying the conditions are outside of the training distribution is vague and unclear.

Please correct mistakes in the paper as I mentioned under "Weaknesses" and conduct a more comprehensive review of recent relighting work to avoid missing important references. This includes the datasets used in Hou et al. 2021 and 2022. 2021 used the DPR dataset and Yale dataset. 2022 used CelebA-HQ: please check this carefully.

---

### Official Review · Reviewer_ST3n · 2024-11-02

**Soundness:** 2
**Presentation:** 3
**Contribution:** 1
**Rating:** 3
**Confidence:** 5

**Summary:**

This paper proposes to more explicitly model the relationship between image and light features, to achieve better single-image face relighting. Overall, the model encodes RGB as well as luminance into image features and (positionally encoded) light properties including XYZ, RGB, and strength into light features, perform cross-attention between them, and finally decode the output relit image.

The authors made a synthetic dataset with Blender in an OLAT setup where they used 7 “maximally separated light colors.” Training losses include RGB reconstruction loss, lighting loss, and a perceptual loss via VGG. The authors also discussed their thougts of not using a SH representation for lighting, as commonly used by other works.

Other technical details that stood out to me include: (1) instead of an MLP, the authors reshaped the embeddings into maps and performed convolution, and (2) they derived KV from lighting features and Q from image features, which is surprising (more below).

The authors show some qualitative and quantitative comparisons against baseline methods plus some ablation studies in the supplemental material PDF.

**Strengths:**

I like the general idea of guiding networks to reason more about the light-face relationship and think it’s essential to eventually enable physically-based relighting with accurate shadows, specular highlights, and even sub-surface scattering.

Since lighting is represented with XYZ, this model is theoretically capable of modeling spatially-varying effects (though no such result was shown, unfortunately).

Writing and presentation are clear, making this paper easy to follow.

**Weaknesses:**

For a relighting paper like this, it’s almost compulsory to provide result videos where a moving light illuminates a face as viewed from a fixed view point. This not only shows how stable/predictable the model is performing for nearby lights but also shows off its theoretical capability of modeling spatially-varying effects.

From the image results presented in the paper, I don’t think the results produced by this method are high-quality. Specifically, the specular highlights are missing/unnatural, and the shading/shadows appear irregular, unlike those cast by natural objects. Admittedly, Table 1 shows this approach achieves the best performance, but Figure 5 shows the proposed method and the baselines are performing, IMO, equally non-pleasing results.

Figure 6, though, shows reasonable results for the middle two columns where the subjects are in front of a clean black background, which resembles the background used by the authors in producing the synthetic dataset for training. This hints at the limitation of training on a synthetic dataset like this, which makes the model struggle with real-world images.

**Questions:**

I am not convinced of the reasons why SH is not desirable here. First, SH is unable to model spatially-varying effects, true, but this paper doesn’t show any of such effects either… Also, the authors claim SH is unable to model RGB lighting, but you can still use SH for strength and RGB as a uniform “scale” applied to the whole SH map. Can the authors clarify if my understanding is correct?

I was expecting KV to be derived from the image features and Q from the lighting features, because intuitively, you want to query with a new lighting condition (via Q) and come up with an answer by "combining" input image patches (via KV), but the authors did this the other way around. What are the intuitions and rationales behind this choice?

---

### Note · Authors · 2024-11-14

**Comment:**

We thank the reviewers for their detailed and thoughtful feedback. We will focus on improving the idea and addressing all the comments.

**Withdrawal Confirmation:**

I have read and agree with the venue's withdrawal policy on behalf of myself and my co-authors.